# A Comparative Analysis of Erythropoietin and Carbamoylated Erythropoietin Proteome Profiles

**DOI:** 10.3390/life11040359

**Published:** 2021-04-19

**Authors:** Neeraj K. Tiwari, Monica Sathyanesan, Vikas Kumar, Samuel S. Newton

**Affiliations:** 1Pediatrics and Rare Disease Group, Sanford Research, Sioux Falls, SD 57104, USA; Neeraj.Tiwari@sanfordhealth.org; 2Division of Basic Biomedical Sciences, Sanford School of Medicine, University of South Dakota, Vermillion, SD 57069, USA; Monica.Sathyanesan@usd.edu; 3Mass Spectrometry and Proteomics Core Facility, University of Nebraska Medical Center, Omaha, NE 68198, USA; Vikas.kumar@unmc.edu

**Keywords:** protein regulation, neurotrophic factors, cognition, hippocampus

## Abstract

In recent years, erythropoietin (EPO) has emerged as a useful neuroprotective and neurotrophic molecule that produces antidepressant and cognitive-enhancing effects in psychiatric disorders. However, EPO robustly induces erythropoiesis and elevates red blood cell counts. Chronic administration is therefore likely to increase blood viscosity and produce adverse effects in non-anemic populations. Carbamoylated erythropoietin (CEPO), a chemically engineered modification of EPO, is non-erythropoietic but retains the neurotrophic and neurotrophic activity of EPO. Blood profile analysis after EPO and CEPO administration showed that CEPO has no effect on red blood cell or platelet counts. We conducted an unbiased, quantitative, mass spectrometry-based proteomics study to comparatively investigate EPO and CEPO-induced protein profiles in neuronal phenotype PC12 cells. Bioinformatics enrichment analysis of the protein expression profiles revealed the upregulation of protein functions related to memory formation such as synaptic plasticity, long term potentiation (LTP), neurotransmitter transport, synaptic vesicle priming, and dendritic spine development. The regulated proteins, with roles in LTP and synaptic plasticity, include calcium/calmodulin-dependent protein kinase type 1 (Camk1), Synaptosomal-Associated Protein, 25 kDa (SNAP-25), Sectretogranin-1 (Chgb), Cortactin (Cttn), Elongation initiation factor 3a (Eif3a) and 60S acidic ribosomal protein P2 (Rplp2). We examined the expression of a subset of regulated proteins, Cortactin, Grb2 and Pleiotrophin, by immunofluorescence analysis in the rat brain. Grb2 was increased in the dentate gyrus by EPO and CEPO. Cortactin was induced by CEPO in the molecular layer, and pleiotrophin was increased in the vasculature by EPO. The results of our study shed light on potential mechanisms whereby EPO and CEPO produce cognitive-enhancing effects in clinical and preclinical studies.

## 1. Introduction

Erythropoietin (EPO) is a 165 amino acid glycoprotein with well-known roles in red blood cell production in the body. Besides erythropoiesis, EPO also acts as an important neurotrophic molecule in brain development. After brain injury, the levels of EPO have been shown to increase in the brain, where it functions as a neuroprotectant [1]. Due to its neurotrophic and neuroprotective effects, EPO has been extensively tested in both preclinical and clinical CNS studies. Treatment-resistant depressed patients treated with EPO in a double-blind, randomized clinical trial reported improvements in depression scores and cognition [2]. EPO clinical trials conducted in combination with brain imaging reported a positive correlation between memory improvement and reversal of brain matter loss in specific hippocampal subregions of depressed patients [3]. These studies indicate that EPO has antidepressant and cognitive-enhancing effects.

Despite the promising results in psychiatric disorders, it is important to note that EPO has potent erythropoietic activity. Chronic administration to non-anemic patients can lead to increased blood viscosity and harmful vascular complications [4]. A chemically engineered modification of EPO, carbamoylated erythropoietin (CEPO), has no erythropoietic effects [5] and produces comparable neurotrophic effects and behavioral effects with EPO [6,7]. Previous studies have shown that ERK signaling can increase the expression of immediate early genes [8]. Most of these immediate early genes are transcription factors that can induce the expression of gene-encoding proteins that control LTP and proteins that can enable new dendritic spine formation. LTP and new dendritic spine formation are primary mechanisms underlying long term memory formation [9,10]. Behavioral studies in mice treated with EPO and CEPO found improved performance in spatial and recognition memory tests, which indicates that both molecules can improve cognition [11]. However, the molecular mechanism involved in their cognitive actions is unknown. The goal of this study was to conduct an unbiased, comparative analysis of EPO and CEPO-induced protein regulation to obtain molecular insight into their mechanism of action. We utilized neuronal phenotype PC-12 cells that have been used extensively to investigate EPO signal transduction [12,13,14]. EPO- and CEPO-treated cell homogenates were subjected to label-free, quantitative proteome analysis. The proteomics data were then subjected to bioinformatics analysis to mine the data for signaling pathways, relationships, and interactions with neurobiological significance. We performed secondary validation of the data using Western blot and immunofluorescence analysis.

## 2. Materials and Methods

### 2.1. Carbamoylation of EPO

Erythropoietin was purchased from Prospec Bio (Ness-Ziona, Israel) and carbamoylated in 1 mg aliquots as previously reported [6,15]. Briefly, EPO was deprotonated in a high pH (pH = 8.9) borate buffer and then exposed to potassium cyanate for 16 h at 36 °C. CEPO was exhaustively dialyzed for 6 h against PBS. CEPO concentration was determined using the Qubit protein assay (ThermoFisher, Waltham, MA, USA). CEPO purity was verified by silver staining after electrophoretic gel analysis.

### 2.2. Cell Culture

Rat pheochromocytoma cells (PC-12 cells) were obtained from the American Type Culture Collection (ATCC). The cells were grown and cultured as mentioned previously with some modifications [16]. The cells were grown in suspension in RPMI-1640 (ATCC) with 10% heat-inactivated horse serum and 5% fetal bovine serum (Gibco, Gaithersburg, MD, USA) at 37 °C and 5% CO_2_. To differentiate the cells into neuronal cells, PC-12 cells were plated in collagen-coated dishes (Corning, Corning, NY, USA) and were grown in RPMI-1640 with NGF (100 ng/mL, Alomone Labs, Jerusalem, Israel) and 1% horse inactivated serum (Gibco). The cells were grown for 10 days and the medium was changed every 2 days. Neuronal morphology and robust neurite outgrowth were confirmed by microscopy. Nerve growth factor (NGF) was removed overnight before the day of experiment. PC-12 cells were treated with EPO and CEPO 100 ng/mL for 5 h. Vehicle-treated (PBS) cells were used as controls. Four replicates were used for each control, and CEPO- and EPO-treated samples were used for label-free quantitative proteome analysis.

### 2.3. Total Protein Extraction

Pellets of 1 × 10^5^ cells from each sample were solubilized in 100 μL 0.1 M Tris–HCl pH 7.6 containing 2% SDS. Each sample was then sonicated for 3 cycles consisting of 15 s of active sonication at 25% amplitude followed by 1 min on ice. Then, the samples were kept at 1 h on a rotator at 4 °C. Samples were then centrifuged for 20 min at 12,000× *g*, and supernatants were collected. Protein concentration in samples was measured by Pierce BCA protein assay (Thermo Fisher Scientific, Rockford, IL, USA). For further downstream analysis, 80 μL of supernatant corresponding to about 100 μg protein was stored at −80 °C.

### 2.4. Label-Free Quantitative Proteome Analysis

A total of 100 µg of protein per sample from four biological replicates per group was taken and detergent was removed by chloroform/methanol extraction, and the protein pellet was re-suspended in 100 mM ammonium bicarbonate and digested with MS-grade trypsin (Pierce, Waltham, MA, USA) overnight at 37 °C. Peptides cleaned with PepClean C18 spin columns (Thermo Scientific™, Waltham, MA, USA) were re-suspended in 2% acetonitrile (ACN), and 0.1% formic acid (FA) and 500 ng of each sample was loaded onto trap column Acclaim PepMap 100 75 µm × 2 cm C18 LC Columns (Thermo Scientific™, Waltham, MA, USA) at a flow rate of 4 µL/min then separated with a Thermo RSLC Ultimate 3000 (Thermo Scientific™, Waltham, MA, USA) on a Thermo Easy-Spray PepMap RSLC C18 75 µm × 50 cm C-182 µm column (Thermo Scientific™, Waltham, MA, USA) with a step gradient of 4–25% solvent B (0.1% FA in 80% ACN) from 10 to 130 min and 25–45% solvent B from 130 to 145 min at 300 nL/min and 50 °C, with a 180 min total run time. Eluted peptides were analyzed by a Thermo Orbitrap Fusion Lumos Tribrid (Thermo Scientific™, Waltham, MA, USA) mass spectrometer in a data-dependent acquisition mode. A survey full scan MS (from *m*/*z* 350 to 1800) was acquired in the Orbitrap with a resolution of 120,000. The AGC target for MS1 was set as 4 × 10^5^ and ion filling time was set as 100 ms. The most intense ions with charge states 2–6 were isolated in 3 s cycles and fragmented using HCD fragmentation with 35% normalized collision energy and detected at a mass resolution of 30,000 at 200 *m*/*z*. The AGC target for MS/MS was set as 5 × 10^4^ and ion filling time was set as 60 ms; dynamic exclusion was set for 30 s with a 10 ppm mass window. Protein identification was performed by searching MS/MS data against the swiss-prot rat protein database downloaded on 13 February 2020 using the in-house mascot 2.6.2 (Matrix Science, Boston, MA, USA) search engine. The search was set up for full tryptic peptides with a maximum of two missed cleavage sites. Acetylation of the protein N-terminus and oxidized methionine were included as variable modifications, and the carbamidomethylation of cysteine was set as a fixed modification. The precursor mass tolerance threshold was set as 10 ppm, and the maximum fragment mass error was 0.02 Da. The significance threshold of the ion score was calculated based on a false discovery rate of ≤1%. Qualitative analyses were performed using progenesis QI proteomics 4.1 (Nonlinear Dynamics).

### 2.5. Bioinformatics and Statistical Analysis

Perseus software (version 1.6.6.0, Max Planck Institute of Biochemistry, Martinsried, Germany) was used to perform bioinformatic and statistical analysis [17]. The normalized LFQ intensities were log2-transformed. Proteins with at least 70% valid values in each group were analyzed. Missing value imputations of protein intensities were performed from a normal distribution (width: 0.3, down shift: 1.8). In order to estimate the variabilities between biological replicates, correlation analyses were performed. After the analysis, one outlier control was removed from the analysis. A column correlation heat map was drawn based on the Pearson correlation coefficients value obtained between biological replicates. In order to estimate the variabilities between biological replicates of the treatment sample, a principal component analysis (PCA) plot was generated with Partek Genomics Suite 7 using protein LFQ values as variables. PCA was performed using logarithmized values without imputation. A multiple-samples test (one-way ANOVA), controlled by a permutation-based FDR threshold of 0.05, was used to identify the significant differences in the protein among Control, EPO and CEPO groups. The logarithmized intensity values of significant proteins from ANOVA after z-score normalization were used for hierarchical clustering using Euclidean distances. The resulting heat map can be interpreted based on color intensity. For enrichment analysis, Fisher’s exact test was computed on gene ontology (GO) terms of significant proteins. Student’s *t*-test was performed for statistical analysis, and statistical filters were set with a *p*-value of 0.05 to detect differential protein ratios between two samples. All those proteins that showed a fold-change of at least ±1.3 and satisfied *p* ≤ 0.05 were considered differentially expressed and were depicted in a Volcano plot. The proteomics data were analyzed using Ingenuity Pathway Analysis software (IPA, Qiagen, Redwood City, CA, USA) to identify the signaling pathways regulated in the study.

### 2.6. Animals

Adult male Sprague Dawley rats (*n* = 6 per group, mass 220–240 gm; Envigo, Indianapolis, IN, USA) were pair-housed according to treatment group (Vehicle, EPO and CEPO) for the duration of the experiments. Rats were maintained on a standard 12 h light–dark cycle with free access to food and water. All procedures were carried out in strict accordance with the National Institutes of Health Guide for the Care and Use of Laboratory Animals, and approval by the USD Institutional Animal Care and Use Committee. Every effort was made to minimize the number of animals used. Rats received single daily i.p. injections of either vehicle (PBS), EPO, or CEPO (30 µg/kg) for 4 consecutive days [6,15]. Five hours after the last dose, animals were decapitated according to American Veterinary Medical Association guidelines. Brain samples were hemisected; one half of the brain was used to dissect out the hippocampus, and the other half was frozen on dry ice and then kept at −80 °C for further use.

### 2.7. Blood Analysis

Adult C57BL/6J mice (*n* = 6) were administered EPO, CEPO (30 µg/kg/day, i.p.) or vehicle (PBS) for a total of 10 doses over 12 days. Trunk blood was collected in Sarstedt lithium heparin tubes (CD300LH). Whole blood samples were analyzed using an IDEXX LaserCyte Dx Hematology Analyzer using the appropriate species-specific settings. Analyses were performed by trained laboratory technicians using two levels of control material.

### 2.8. Western Blot Analysis

Western blot analysis was used to quantify changes in phospho-signaling proteins [18]. The hippocampus samples were homogenized in the RIPA buffer with the complete protease inhibitor cocktail (ThermoFisher, Waltham, MA, USA). Homogenates (30 μg) were mixed with Laemmli sample buffer and then resolved by SDS-PAGE using 5–14% at 60 V for 30 min followed by 90 V for 2 h. Proteins were blotted to nitrocellulose membranes, which were blocked by 1% BSA in Tris-buffered saline (TBS; 25 mM Tris–HCl, pH 7.4, 0.9% NaCl) containing 0.1% Tween 20 (TBS-T) for 1 h at room temperature, and then probed overnight at 4 °C with primary antibodies diluted in TBS-T. Primary antibodies were Phospho AKT (ser473) (Cell Sig, 193H12, 1:1000 dilution), Phospho-p44/42 MAPK (Erk1/2) (Cell Signaling, 4370S, 1:1000 dilution), and glyceraldehyde 3-phosphate dehydrogenase (GAPDH) (Cell Signaling, Danvers, MA, USA, 5174S, 1:1000 dilution), which was used as a loading control. Membranes were rinsed and incubated with secondary antibodies for 1 h at room temperature. Secondary antibody AF680 goat anti-rabbit (ThermoFisher, Waltham, MA, USA, A21109, 1:1000 dilution) was used to visualize the detected proteins with the Odyssey Infrared Imaging System. Semi-quantitative analysis was performed by Image Studio Lite 5.2.5. ANOVA was calculated using GraphPad-Prism. A difference was considered as significant when the *p*-value was less than 0.05 (*p* < 0.05). Data were reported as mean ± SEM.

### 2.9. Immuno-Fluorescence Analysis

Immuno-fluorescence analysis on hemisected rat brains was performed as previously described [19]. Briefly, 16 µm coronal, cryocut sections were incubated overnight at 4 °C with primary antibody (Grb2, Santa Cruz Biotechnology, 1:200; Cortactin, Thermofisher, Waltham, MA, USA, 1:500; Pleiotrophin, 1:100) in antibody solution. Antibodies were used as per the manufacturer’s instructions, and specificity was tested using incubation in antibody solutions lacking primary antibody. Following primary antibody incubation, slides were rinsed in PBS and then incubated in fluorescent secondary antibody (Alexa-594, Alexa-488 1:500) for 2 h at room temperature. Slides were then rinsed in PBS and coverslips secured using VectaMount (Vector Labs, Burlingame, CA, USA). Sections were viewed and images captured using a Nikon Eclipse Ni microscope equipped with a DS-Qi1 monochrome, cooled digital camera, and NIS-AR 4.20 Element imaging software. Sections from EPO-, CEPO- and vehicle (PBS)-treated rat brain sections were captured using identical exposure settings.

## 3. Results

### 3.1. Hematological Analysis

Several hematological parameters were measured after EPO and CEPO administration (Table 1). EPO strongly elevated several of the measured values, while CEPO was comparable to control values. EPO increased red blood cell (RBC) counts by 60% and drastically increased the reticulocyte number, which was nine-fold higher than the control. EPO also doubled platelet counts, whereas platelet counts in the blood of CEPO-treated animals were similar to the control.

Whole blood from EPO- and CEPO-treated mice was analyzed using an IDEXX hematology analyzer. Mice were administered 10 doses of EPO or CEPO (30 µg/kg/day) in PBS. Data shown are mean values from *n* = 6 mice.

### 3.2. Exploratory Analysis of the LFQ Data for EPO- and CEPO-Treated Neuronal Cell Cultures

A total of 2216 proteins were identified, and 2121 proteins were quantified with at least 70% valid values in each group. The reproducibility of the biological replicates was assessed by the column correlation heatmap (Figure 1A). The hierarchical cluster for the column correlation was derived by the Pearson correlation coefficient (PCC) values determined based on label-free quantitation (LFQ) intensities (Appendix A). The heatmap shows a very high correlation between replicates of the same treatment.

Data indicate that the sample replicates had a high degree of reproducibility. All quantified proteins were explored by principal component analysis (PCA), which displayed three different clusters according to their abundance variation (Figure 1B). Principal component 1, which consisted of 29.5% of the total variation, and principal component 2, which consists of 17.9% of variation, led to separation of the control, EPO and CEPO samples into different principal components. The close clustering of samples within the groups indicates high consistency of Control, EPO and CEPO samples.

### 3.3. Enrichment Analysis of the LFQ Data for EPO- and CEPO-Treated Neuronal Cell Cultures

There were 605 significant proteins out of 2121 proteins after ANOVA analysis (Appendix A). Hierarchical clustering analysis (HCA) of the significant proteins was performed to identify the protein groups with a similar expression pattern (Figure 2A). The biological replicates of each treatment condition represent the column header. The cluster analysis also shows different regulation patterns for different groups of proteins based on treatment. The expression of proteins is either up or down regulated or remains unchanged. The HCA grouped all the significant proteins into seven main clusters. Cluster 1 consists of 140 proteins, cluster 2 consists of 15 proteins, cluster 3 consists of 49 proteins, cluster 4 consists of 45 proteins, cluster 5 consists of 26 proteins, cluster 6 consists of 278 proteins, and cluster 7 consists of 134 proteins. The profile plot for cluster 1 indicates a protein group that is upregulated with both EPO and CEPO treatment, cluster 3 indicates a protein group that is upregulated with CEPO treatment, and cluster 7 indicates a protein group that is upregulated with EPO (Figure 2B). We conducted enrichment analyses to find significant physiological functions regulated by these proteins. Using Fisher’s exact test, the enrichment analysis of different protein clusters for gene ontologies and pathways was performed (Figure 2C and Appendix A). We found that cluster 3 was enriched for the function related to CNS myelination, astrocyte development, regulation of neurogenesis, axon development, and activating transcription factor binding. Cluster 7 was enriched for the function related to the regulation of developmental growth, endothelial cell proliferation, positive regulation of LTP, ionotropic glutamate receptor binding, and the positive regulation of axon extension. Cluster 1 was enriched for functions related to the regulation of neurogenesis, neurotransmitter transport, regulation of synaptic plasticity, memory, and neurotrophic signaling.

### 3.4. Signaling Pathways Upregulated with EPO and CEPO Treatment

Ingenuity pathway analysis (IPA) was used to identify canonical signaling pathways that were significantly upregulated in EPO vs. Control and CEPO vs. Control (Figure 3A,B, Appendix A). There are many canonical signaling pathways related to memory formation that were significantly upregulated. There was significant increase in ERK/MAPK signaling with both EPO (−log *p*-value = 7.08) and CEPO (−log *p*-value = 7.08). There was significant increase in CREB signaling with both EPO (−log *p*-value = 3.26) and CEPO (−log *p*-value = 3.26). There was significant increase in synaptic long-term potentiation signaling with both EPO (−log *p*-value = 7.36) and CEPO (−log *p*-value = 7.36). There was significant increase in synaptogenesis signaling with both EPO (−log *p*-value = 9.10) and CEPO (−log *p*-value = 9.10). EPO- and CEPO-treated rat hippocampal samples were used for Western blot studies to further confirm the activation of PI3/AKT signaling and ERK/MAPK signaling results from the IPA analysis. We found a significant increase in the phospho-AKT signaling molecule both in EPO- and CEPO-treated rat hippocampal samples (Figure 4A,B). Additionally, we found a significant increase in the phospho-ERK1/2 signaling molecule both in EPO- and CEPO-treated rat hippocampal samples. (Figure 4C,D).

### 3.5. Differentially Expressed Proteins in EPO- and CEPO-Treated Neuronal Cell Cultures

In EPO vs. Control, 136 differentially regulated proteins out of 2121 identified proteins showed significant *p*-values after applying Student’s *t*-test. The significant proteins were plotted for the *p*-values and *t*-test difference. The *t*-test difference was set as ±0.379 to obtain significant differentially expressed proteins.

Differentially expressed proteins were those with a ±1.3-fold change. A total of 104 proteins had a ≥1.3-fold increased expression, whereas 18 proteins demonstrated a ≤1.3-fold decreased expression (Figure 5A). For CEPO vs. Control, 266 proteins out of 2121 proteins showed significant *p*-values after applying the Student’s *t*-test. The significant proteins were plotted for the *p*-values and *t*-test differences. The *t*-test difference was set as ±0.379 to obtain differentially expressed proteins with a ±1.3-fold change. Eighty-five proteins had a ≥1.3-fold increased expression, whereas 148 proteins had a ≤1.3-fold decreased expression (Figure 5B). Among the differentially expressed proteins, synaptic proteins such as synaptosomal-associated protein, 25 kDa (SNAP-25), Sectretogranin-1 (Chgb), Cortactin (Cttn), calcium/calmodulin-dependent protein kinase type 1 (Camk1), elongation initiation factor 3a (Eif3a), 60S acidic ribosomal protein P2 (Rplp2) that have roles in synaptic plasticity and cognition were upregulated with both EPO and CEPO treatment compared to the control (Appendix A). For EPO vs. CEPO, 691 proteins out of 2121 proteins showed significant *p*-values after applying the Student’s *t*-test. The significant proteins were plotted for the *p*-values and *t*-test differences. The *t*-test difference was set as ±0.379 to obtain differentially expressed proteins with a ±1.3-fold change. A total of 443 proteins were expressed at higher levels in EPO-treated cells as compared to CEPO-treated cells, and 101 proteins were expressed at lower levels, using the cutoffs outlined above (Figure 5C). Trophic factor proteins such as Neudesin (Nenf), Pleiotrophin (PTN), Myotrophin (Mtpn), and proteins related to erythropoiesis such as Hemoglobin subunit beta (HBB) were upregulated with EPO treatment only (Appendix A).

### 3.6. Immuno-Fluorescence Analysis in Brain Tissue

Qualitative analysis of in vivo brain expression of EPO- and CEPO-induced proteins was performed by immuno-fluorescence analysis using commercially available antibodies (Figure 6). Although hippocampal sections were used, we examined the entire section for differential protein expression between the three experimental groups. Brain subregions exhibiting the highest differential regulation are shown. Growth factor receptor-bound 2 (Grb2) expression was increased by both EPO and CEPO administration, specifically in the dentate gyrus (Figure 6A). Cortactin was elevated only by CEPO and was most noticeable in the dentate gyrus molecular layer (Figure 6B). Pleiotrophin expression was detected at low levels in cortical vasculature and was elevated only by EPO (Figure 6C). Vascular cell phenotype was determined by morphology of staining (dotted ovals, Figure 6C).

## 4. Discussion

The neuroprotective and neurotrophic actions of EPO have made it a useful molecule to investigate in clinical studies of neuropsychiatric disorders. As an FDA-approved biologic drug that is widely prescribed to treat anemia, the safety profile is well documented. However, the potential for adverse hematological effects with chronic dosing is a major limitation for its use as a CNS drug. CEPO is devoid of erythropoietic activity and helps to address this key limitation [5]. Our results also show that CEPO had no effect on platelet counts, whereas EPO sharply elevated it. EPO increased the reticulocyte number nine-fold. CEPO did not increase the reticulocyte number over control levels. The apparent lack of hematopoietic cascade activation raises interesting questions regarding CEPO’s mechanisms of action in mediating behavioral effects comparable to EPO [7,20].

Our enrichment analysis of EPO- and CEPO-induced proteins and their respective functions provides additional insight into their potential mechanisms of action. Both ligands elevated the expression of proteins that regulate functions such as neurogenesis, neurotransmitter transport, regulation of synaptic plasticity, memory, and neurotrophic signaling. This correlates to the EPO and CEPO gene expression study, where genes related to neurogenesis, neurotransmitter transport, and synaptic plasticity were upregulated [6]. Gene expression studies have shown that EPO [21] and CEPO [15] share an overlap in their neurotrophic factors, such as BDNF, VGF and neuritin, and immediate early gens such as Arc, fos and Egr1 [6]. These neurotrophic molecules produce antidepressant effects, and by activating immediate early genes induce synaptic plasticity, LTP, and dendritic spine development that can have cognitive-enhancing effects [21,22,23,24,25,26,27,28].

Both molecules comparably induced MAPK and Akt in the rat hippocampus. These cascades could be involved in their behavioral effects because these kinase pathways have been strongly implicated in antidepressant-like activity as well as memory formation by inducing synaptic plasticity [29,30,31]. It is tempting to speculate that CEPO [7] recapitulates EPO’s antidepressant [2,3] effects by virtue of activating trophic signaling pathways but is non-erythropoietic because it does not induce the canonical Jak-STAT hematopoietic cascade [5]. While the results from this study indicate an overlap in trophic pathways, we did not find evidence indicating selective activation of the hematopoietic pathway by EPO. Previous work that carefully examined the differences in hematopoietic signaling molecules induced by wildtype EPO and a non-erythropoietic mutant EPO reported differences that were subtle and dynamic [32]. Our studies, conducted at a single timepoint, were likely unable to capture these dynamic changes. A global phospho-proteome approach capturing changes shortly after receptor activation has the potential to shed light on dynamic and differential signaling pathway activation by EPO and CEPO. 

Bioinformatics pathway analysis revealed both EPO and CEPO induced proteins with functions related to neurogenesis, synaptic plasticity, neurotransmitter transport, synaptic vesicle priming, LTP and dendritic spine development (Figure 7). Upregulated proteins included synaptosomal-associated protein, 25 kDa (SNAP-25), Sectretogranin-1 (Chgb), Cortactin (Cttn), calcium/calmodulin-dependent protein kinase type 1 (Camk1), elongation initiation factor 3a (Eif3a), and 60S acidic ribosomal protein P2 (Rplp2), which regulated the release of the presynaptic vesicle and thus long-term potentiation and synaptic plasticity in the hippocampus [33]. A secretory protein present in synaptic vesicles, Chgb, promotes neurotransmitter release and differentiation of the hippocampal neuronal precursor cells [34,35]. An F-actin binding protein, Cttn, is present in dendritic spines in the hippocampus. During the synaptic activity, it causes changes in spine shape and size by interacting with actin filaments and supporting the induction of LTP. Additionally, Cttn interacts with PSD-95, causing an increase in spine density and facilitates LTP and synaptic plasticity [36,37]. CEPO induced Cortactin specifically in the molecular layer of the dentate gyrus, which could indicate that CEPO’s actions prominently involve the hippocampus. Long-term memory formation occurs due to an increase in synaptic strength and is facilitated by new protein synthesis. The key components of protein synthesis are elongation factors such as Eif3a that aid in the protein synthesis initiation step and ribosomal subunits, such as Rplp2, involved in protein translation [38,39,40].

Our comparative analysis of EPO- and CEPO-induced protein expression profiles provides additional insight into their potential mechanisms of action. Both ligands elevated the expression of neurotrophic and neurogenic proteins. Interestingly, more classes of trophic factor molecules were induced by EPO than CEPO. Trophic factor proteins such as Neudesin (Nenf), Pleiotrophin (PTN), Myotrophin (Mtpn), and proteins related to erythropoiesis such as hemoglobin subunit beta (HBB) were upregulated with EPO treatment only. This suggests that CEPO has a more limited trophic role in comparison to EPO, which is known to be pleiotrophic. It is likely that this is due to differential activation of intracellular signal transduction cascades by EPO and CEPO. 

The use of PC12 cells has limitations because it is a cell line and not a direct representation of brain tissue. EPO and CEPO effects are likely to involve actions on multiple cell types, including neurons, endothelial cells, and astrocytes. In order to obtain a high-resolution comparative analysis of EPO- and CEPO-induced proteomes in neuronal cells, we used differentiated, neuronal morphology PC12 cells. We confirmed key signaling pathways using hippocampal tissue and in vivo protein expression using immunofluorescence analysis.

Overall, the EPO- and CEPO-induced protein expression profiles provide mechanistic insight into their behavioral actions; particularly, the cognitive effects that have been reported in preclinical and clinical studies [3]. Our study was focused on global protein expression changes that are essentially downstream from receptor activation and did not capture alterations that are transient and dynamic. In future studies aimed at understanding CEPO’s lack of hematopoietic effects, it will be useful to focus on post-translational modifications that regulate signal transduction. It is widely thought that CEPO signals via a beta common receptor and EPO receptor heteromer rather than the EPO receptor dimer employed by EPO [42]. Further studies are needed to understand this important ligand–receptor interaction and how it affects cellular signaling. The possibility of additional receptors and adaptor molecules should also be considered. 

## Figures and Tables

**Figure 1 life-11-00359-f001:**
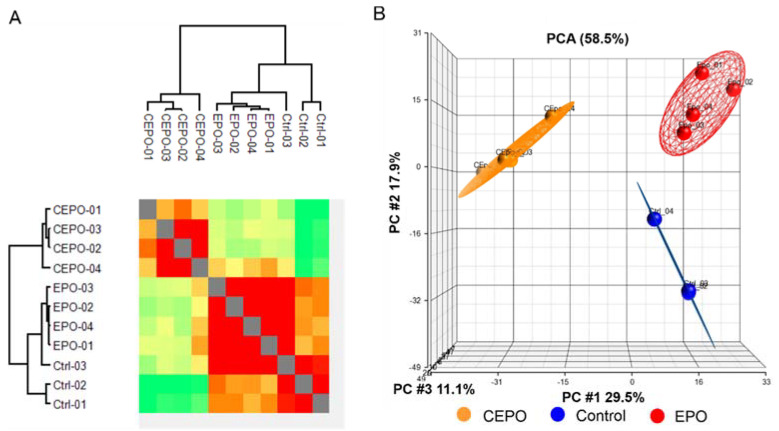
Exploratory analysis of the LFQ data for EPO- and CEPO-treated neuronal cell cultures. (**A**) Hierarchical clustering of all samples is based on Pearson correlation coefficients. Correlation values were color-coded from blue to red, corresponding to lower or higher values. (**B**) Principal component analysis (PCA) of the LFQ intensities obtained from the control, EPO-, and CEPO-treated samples.

**Figure 2 life-11-00359-f002:**
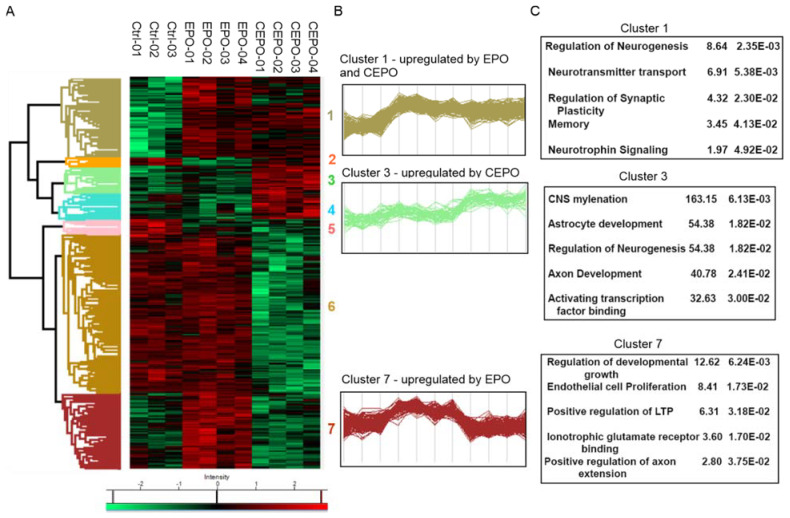
Enrichment analysis of the LFQ data for EPO- and CEPO-treated neuronal cell cultures. (**A**) Heatmap of the significant proteins expressed in CEPO, EPO and Control samples. Hierarchical clustering was performed for tables, where rows represent one protein, and columns represent biological replicates. Significant proteins were calculated with multi-sample ANOVA tests with a permutation-based cutoff of 0.05 applied on the logarithmic intensities. Intensity values were color-coded from green to red, corresponding to downregulation or upregulation, respectively. (**B**) Profile plot for three selected clusters showing distinct behavior with respect to different treatments includes Cluster 1, strongly expressed with both EPO and CEPO treatment; Cluster 3, strongly expressed with CEPO treatment; and Cluster 7, strongly expressed with EPO treatment. (**C**) Enrichment analysis of protein annotations shows functional categories enriched in the three selected clusters, 1, 3, and 7. The enriched terms, the corresponding enrichment factor, and *p*-value are shown.

**Figure 3 life-11-00359-f003:**
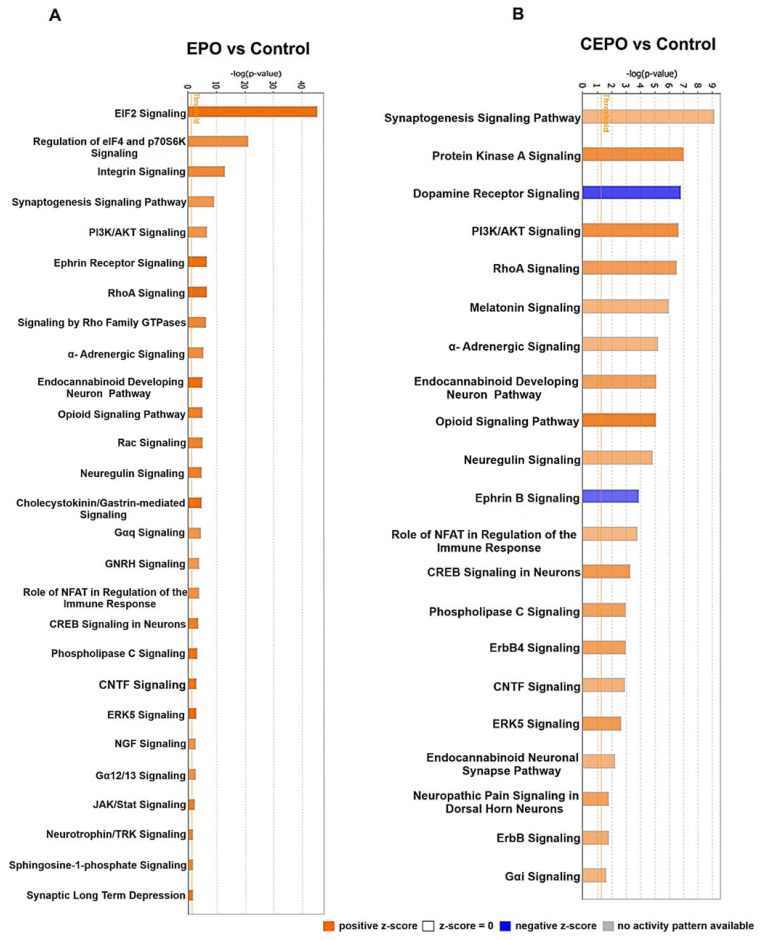
The signaling pathways upregulated in EPO- and CEPO-treated neuronal cell cultures. (**A**) EPO vs. Control. (**B**) CEPO vs. Control. The canonical pathways are represented on the *x*-axis. The *y*-axis represents the significance scores as −log *p*-value. The threshold line indicates the significance (*p* < 0.05) cutoff. The height of the bar shows the level of significance.

**Figure 4 life-11-00359-f004:**
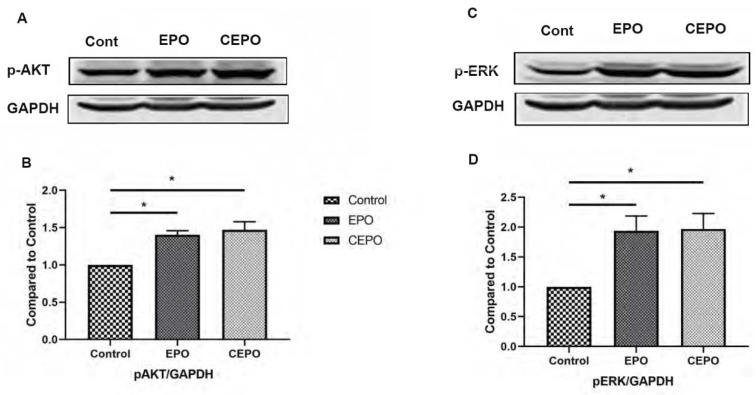
EPO and CEPO treatment upregulates pAKT and pERK signaling in rat hippocampal samples. (**A**) Western blot. (**B**) Graphical representation of Western blot results showing an increase in phosphorylated-AKT in the hippocampus after 4 days of EPO and CEPO (30 µg/kg) treatment in Sprague Dawley rats (*n* = 6). (**C**) Western blot. (**D**) Graphical representation of Western blot results showing an increase in phosphorylated-ERK1/2 in hippocampus after 4 days of EPO and CEPO (30 µg/kg) treatment in Sprague Dawley rats (*n* = 6). The data are reported as mean (±SEM) and *p*-values < 0.05 (*****) were considered as significant.

**Figure 5 life-11-00359-f005:**
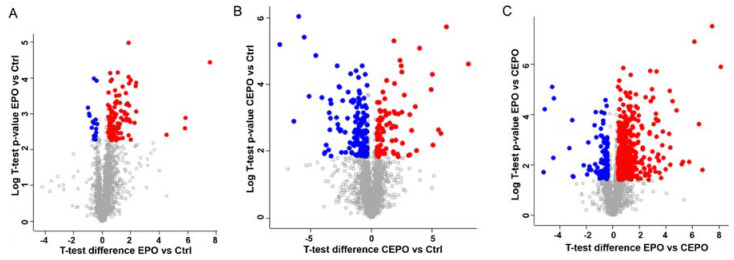
Differentially expressed proteins in EPO- and CEPO-treated neuronal cell cultures. Volcano plots of differentially expressed proteins between the experimental groups are shown. (**A**) EPO vs. Control: 104 were upregulated and 18 were downregulated in EPO as compared to the control. (**B**) CEPO vs. Control: 85 proteins were upregulated and 148 were downregulated in CEPO as compared to the control. (**C**) EPO vs. CEPO: 443 proteins were upregulated and 101 proteins were downregulated in EPO as compared to CEPO. For the graph, −log (*p*-value) is plotted against the *t*-test difference. The downregulated proteins are on the left and significant ones are in blue; the upregulated proteins are on the right and significant ones are in red. The cutoff value for differentially expressed proteins was set at ±1.3-fold (0.379 in log2-transformed values).

**Figure 6 life-11-00359-f006:**
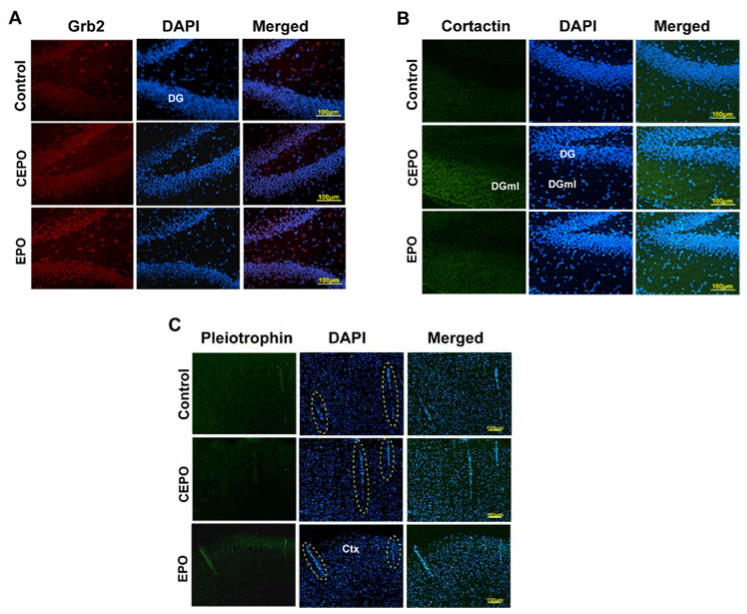
Immuno-fluorescence analysis of protein expression in the rat brain. Rats were administered EPO or CEPO for 4 days (30 µg/kg/day). Cryocut hippocampal brain sections were processed for immunofluorescence detection of 3 proteins in the 3 experimental groups, EPO, CEPO and Control (PBS). Representative images are shown from *n* = 4 analyses. (**A**) Growth factor-bound 2 (Grb2); (**B**) Cortactin; and (**C**) Pleiotrophin. Dotted ovals indicate vasculature in the cortex. DG, dentate gyrus; DGml, dentate gyrus molecular layer; Ctx, cortex.

**Figure 7 life-11-00359-f007:**
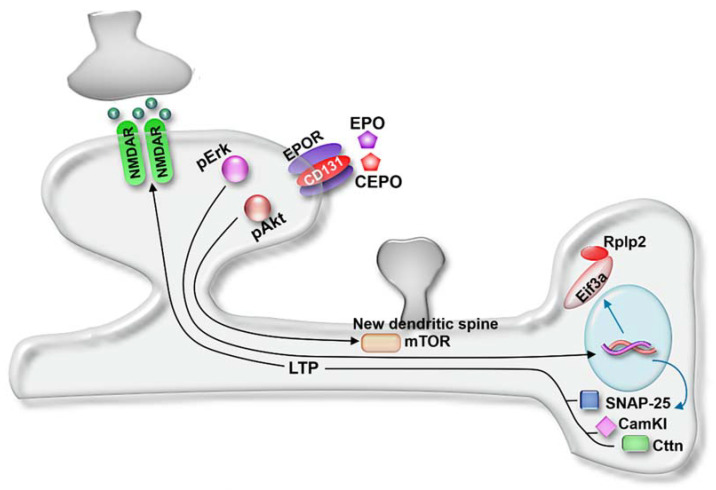
Model of EPO and CEPO actions. Molecules and signaling pathways regulated by EPO and CEPO were integrated to develop a mechanistic model involving synapse activity, LTP, and spine generation. NMDAR, glutamate receptor; CD131, beta common receptor; mTOR, mammalian target of rapamycin; Eif3a, elongation initiation factor 3a; Rplp2, 60S acidic ribosomal protein P2; CamKI, calcium/calmodulin-dependent protein kinase I; Cttn, cortactin; LTP, long-term potentiation. Model was adapted from [41].

**Table 1 life-11-00359-t001:** Comparative analysis of hematological parameters.

Hematological Parameters	Veh	EPO	CEPO
RBC (M/uL)	10.26	16.49	10.48
Hemoglobin (g/dL)	15.64	23.00	16.07
Hematocrit (%)	48.78	84.55	49.37
Mean corpuscular volume (fL)	46.80	51.30	46.95
Mean corpuscular Hb conc (g/dL)	33.08	27.22	32.78
Red cell distribution width (%)	24.78	38.98	25.78
Reticulocyte number (K/uL)	442.92	3916.62	475.02
Reticulocyte percent	4.37	23.76	4.47
Platelet count (K/uL)	152.20	366.67	151.83

## Data Availability

The datasets analyzed during the current study are available from the corresponding author on reasonable request.

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
