# Peer review of "A Comparative Analysis of Erythropoietin and Carbamoylated Erythropoietin Proteome Profiles"

_life, 2021, doi:10.3390/life11040359_

Round 1

Reviewer 1 Report

Carbamylated erythropoietin (cEpo), with neuroprotective but without hematopoietic activity, has been attracting rising concerns. In the current study the authors have tried to provide additional insight and shed new light into the potential mechanisms of action of erythropoietin (EPO and CEPO in regards to their antidepressant activity. The authors have performed an interesting bioinformatic analysis of proteins involved in the signaling pathways of the above compounds and they have examined relationships and interactions. The study is interesting but because of the many protens. etc, involved in the EPO/CEPO pathways the study is not easily understood by scientists of other fields and thus a scheme showing the connection of the factors examined would be very helpful. Comments 1. Introduction: The authors should add extra information in regards to multiple proteins/pathways examined 2. Animals/ Methodology a)Why have the authors used 2 different kinds of experimental animals? b) Why have the authors used 2 different therapeutic protocols i.e 4 and 5 days of EPO/ CEPO administration? c) Upon which criteria have the authors selected the dosage (dose and time) of the compounds used for the treatment of PC12 cells and the animals. Have they tried other doses? Is there any correlation to the doses that have been administered to patients? 3.Results: a)a scheme illustrating the pathways and their connection would be of great help. b) the 3 clusters give important information but a table of the EPO/CEPO functions and the relative proteins/ and the relative function referred is necessary 4. Discussion: The information is abundant and this section needs to be enriched with some extra comments. The last phrase is better to be commented in previous paragraphs. 5. Are there any limitations of the study? 6 Lines 244-246: The phrases do not seem connected

Author Response

Reviewer 1

Carbamoylated erythropoietin (cEpo), with neuroprotective but without hematopoietic activity, has been attracting rising concerns. In the current study the authors have tried to provide additional insight and shed new light into the potential mechanisms of action of erythropoietin (EPO and CEPO in regard to their antidepressant activity. The authors have performed an interesting bioinformatic analysis of proteins involved in the signaling pathways of the above compounds and they have examined relationships and interactions. The study is interesting but because of the many protens. etc, involved in the EPO/CEPO pathways the study is not easily understood by scientists of other fields and thus a scheme showing the connection of the factors examined would be very helpful.

Response: We have added a model of EPO/CEPO actions “image-7” illustrating the regulated molecules and their connection.  

Comments 1. Introduction: The authors should add extra information regarding multiple proteins/pathways examined

Response: Extra information regarding multiple proteins/pathways have been added to the introduction from line number 52 to line number 57. “Previous studies have shown that ERK signaling can increase the expression of immediate early genes. Most of these immediate early genes are transcription factors that can induce expression of genes encoding proteins that controls LTP and also proteins that can enable new dendritic spine formation. LTP and new dendritic spine formation are the primary mechanism underlying long term memory formation.”

  1. Animals/ Methodology a) Why have the authors used 2 different kinds of experimental animals? b) Why have the authors used 2 different therapeutic protocols i.e 4 and 5 days of EPO/ CEPO administration? c) Upon which criteria have the authors selected the dosage (dose and time) of the compounds used for the treatment of PC12 cells and the animals. Have they tried other doses? Is there any correlation to the doses that have been administered to patients?

Response: a) We would like to clarify that PC12 cells are of rat origin and the in vivo studies, western blot (Fig. 4) and immunofluorescence (Fig. 6) used rat brain sections. Mouse was used for the blood profile due to the availability of a standardized assay for mouse blood profile in the Idexx instrument. b) The EPO/CEO dosing regimen was 4 days. Animals were sacrificed 5 hours after the last dose. c) dose for PC12 cells and rats were selected from previous studies of gene profiling, Tiwari et al, Prog Neuropsychopharmacol Biol Psychiatry. 2019 Jan 10;88:132-141, and modulation of cognition in rat social defeat stress, Sathyanesan et al, Transl Psychiatry. 2018 Jun 8;8(1):113. As this dose has consistently produced robust behavioral effects in antidepressant response and cognitive behavioral assays, we continue to study molecular effects of the behaviorally active dose. Humans studies, limited to EPO, have employed 50,000 – 100,000 units/dose and dosed once a week for multiple weeks. One unit of EPO corresponds to approximately 8 nanograms.    

  1. Results: a) a scheme illustrating the pathways and their connection would be of great help. b) the 3 clusters give important information but a table of the EPO/CEPO functions and the relative proteins/ and the relative function referred is necessary.

Response: a) We added a mechanistic model “image-7” illustrating the pathways and their connection.

  1. b) Section c. of image 2 shows the functions enriched with EPO/CEPO treatment. We are modifying the image so that it will be easily visible the functions that are enriched with EPO/CEPO treatment. Details of the functions and proteins enriched with the EPO/CEPO treatment is provided in the supplementary files Supplementary table S3.
  2. Discussion: The information is abundant, and this section needs to be enriched with some extra comments. The last phrase is better to be commented in previous paragraphs.

Response: We have moved the last phrase to the end of the third paragraph of the Discussion. A paragraph on the limitations of the study has been included towards the end of the Discussion, lines 441 – 447.

  1. Are there any limitations of the study? 6 Lines 244-246: The phrases do not seem connected.

Response: The use of PC12 cells is a limitation because it is a cell line and not an in vivo study. However, we tested a subset of regulated candidates and key pathways in rat brain. A paragraph has been added towards the end of the Discussion to explain limitations. Lines 252-255 have been rephrased “Profile plot for three selected clusters showing distinct behavior with respect to different treatments includes Cluster 1 strongly expressed with both EPO and CEPO treatment, Cluster 3 strongly expressed with CEPO treatment and Cluster 7 strongly expressed with EPO treatment.”

Reviewer 2 Report

The paper is valuable because exhaustively comparing with former works describes the efects of CEpo on culture cells and mice brain. I hope it will help in development of use of Epo devoid of hematological action in depression treatment.

Author Response

We are glad that the reviewer found our work interesting and timely.

Reviewer 3 Report

This is a well-conducted study that provides an important contribution to our understanding of the effect of epo and cepo on protein expression. The methods are adequately described and the paper is generally well written. The results are interpreted cautiously and appropriately.

I congratulate the authors on this work.

I have a few suggestions that might improve the manuscript.

Major comments:

The markers analyzed by immunofluorescence should be mentioned in the abstract, including what the results were.

In the mouse study epo/cepo was administered for longer than in the rat study. Was the time of administration in the rat study sufficient? Can you discuss this?

Discuss the limitations of using PC12 cells rather than whole animals for the proteomics study.

Minor comments:

Double check for definition of all abbreviations. Are they necessary? If a term only appears two or three times in the manuscript, you can just as well spell it out.

Line 214, “these” before figure 1?

Line 233 “The HCA Figure 2.” Is not a complete sentence.

Line 280 “Ingenuity Pathway Analysis (IPA) was used toFigure 3. The…” the formatting of the manuscript is messed up. The text does not wrap correctly when figures and figure legends are inserted.

Figure legend of figure 5 is very repetitive; can this be said more concisely?

Ad “Differentially expressed proteins in EPO and CEPO treated neuronal cell culture” This section is very difficult to read. Can you write it more concisely, telling about cutoffs that are used for all comparisons only once? Please emphasize which proteins are up- or down regulated in which comparisons.

Line 315 “Among the differentially expressed proteins, synaptic proteins such as SNAP25, Chgb, Cttn, Camk1, Eif3a, Rplp2 were upregulated that have a role in synaptic plasticity and cognition that were upregulated with both EPO and CEPO treatment” this sentence needs revision. “compared to control”

Line 321 “Four hundred forty-three proteins had ≥ 1.3-fold increase expression whereas 101 proteins had ≤ -321 1.3-fold decreased expression (Fig 5C).” increased in epo vs cepo or the other way round? You can just say, “443 proteins were expressed at higher levels in cepo treated cells as compare to epo treated cells and 101 proteins were expressed at lower levels, using the cutoffs outlined above”

Line 349: “EPO’s effect on reticulocyte number 9 times higher than control.” What do you mean?

Line 350: “As with other hematological parameters, CEPO’s effect on reticulocyte number was comparable to controls.” This is a strange way of saying that CEPO did not increase reticulocyte number over control levels. Consider revising.

Author Response

Reviewer 3

This is a well-conducted study that provides an important contribution to our understanding of the effect of epo and cepo on protein expression. The methods are adequately described and the paper is generally well written. The results are interpreted cautiously and appropriately.

I congratulate the authors on this work.

I have a few suggestions that might improve the manuscript.

Major comments:

  1. The markers analyzed by immunofluorescence should be mentioned in the abstract, including what the results were.

Response: We included the immunofluorescence information and results in the abstract.

  1. In the mouse study epo/cepo was administered for longer than in the rat study. Was the time of administration in the rat study sufficient? Can you discuss this?

Response: Our goal in mouse blood profile analysis was to rigorously test CEPO for hematological effects in parallel with EPO. EPO is unlikely to produce an increase in hematocrit after only 4 doses, and hence the decision to employ an extended regimen that strongly elevated hematocrit. We have previously shown, Sathyanesan et al., Transl Psychiatry. 2018 Jun 8;8(1):113, in a rat model of social defeat stress, that 4 doses of CEPO reverse cognitive deficits in behavioral assays and increase neurotrophic gene expression in the hippocampus. This indicates that 4 doses of CEPO are sufficient to produce procognitive behavioral effects. 

Discuss the limitations of using PC12 cells rather than whole animals for the proteomics study.

Response: We acknowledge that the use of PC12 cells to investigate EPO and CEPO proteome profiles does involve some limitations because it is a cell line and not a direct representation of brain tissue. EPO and CEPO effects are likely to involve actions on multiple cell types, including neurons, endothelial cells and astrocytes. In this context, experiments utilizing whole brain regions can yield mixed data. In order to obtain a high-resolution comparative analysis of EPO and CEPO-induced proteomes in neuronal cells, we used differentiated, neuronal morphology PC12 cells. However, we confirmed key signaling pathways using hippocampal tissue and in vivo protein expression using immunofluorescence analysis.

Minor comments:

Double check for definition of all abbreviations. Are they necessary? If a term only appears two or three times in the manuscript, you can just as well spell it out.

Response: Changes have been made line 451-454 also line 518-520.

Line 214, “these” before figure 1?

Response: Formatting error problem has been fixed Line 226.

Line 233 “The HCA Figure 2.” Is not a complete sentence.

Response: Formatting error problem has been fixed Line 246.

Line 280 “Ingenuity Pathway Analysis (IPA) was used to Figure 3. The…” the formatting of the manuscript is messed up. The text does not wrap correctly when figures and figure legends are inserted.

Response: Formatting error problem has been fixed in Line 297.

Figure legend of figure 5 is very repetitive; can this be said more concisely?

Ad “Differentially expressed proteins in EPO and CEPO treated neuronal cell culture” This section is very difficult to read. Can you write it more concisely, telling about cutoffs that are used for all comparisons only once? Please emphasize which proteins are up- or down regulated in which comparisons.

Response: The legend for Figure 5 has been reworded, lines 322 - 327.

Line 315 “Among the differentially expressed proteins, synaptic proteins such as SNAP25, Chgb, Cttn, Camk1, Eif3a, Rplp2 were upregulated that have a role in synaptic plasticity and cognition that were upregulated with both EPO and CEPO treatment” this sentence needs revision. “compared to control”

Response:  Sentence has been revised, corrected Lines 335-339.

Line 321 “Four hundred forty-three proteins had ≥ 1.3-fold increase expression whereas 101 proteins had ≤ -321 1.3-fold decreased expression (Fig 5C).” increased in epo vs cepo or the other way round? You can just say, “443 proteins were expressed at higher levels in cepo treated cells as compare to epo treated cells and 101 proteins were expressed at lower levels, using the cutoffs outlined above”

Response: Sentence corrected in Lines 343-344.

Line 349: “EPO’s effect on reticulocyte number 9 times higher than control.” What do you mean?

Response:  Sentence now reads “EPO increased reticulocyte number 9-fold.

Line 350: “As with other hematological parameters, CEPO’s effect on reticulocyte number was comparable to controls.” This is a strange way of saying that CEPO did not increase reticulocyte number over control levels. Consider revising.

Response: Sentence now reads “CEPO did not increase reticulocyte number over control levels”.

Round 2

Reviewer 1 Report

The authors have successfully replied to the comments. The paper has been significantly improved and the scheme is helpful.. Minor editing is needed.